# Change in Neuroticism and Extraversion among Pre-University Education Employees during the COVID-19 Pandemic

**DOI:** 10.3390/medicina58070895

**Published:** 2022-07-04

**Authors:** Teodora-Gabriela Alexescu, Mădălina-Stela Nechita, Anca-Diana Maierean, Damiana-Maria Vulturar, Mircea Ioan Handru, Daniel-Corneliu Leucuța, Olga Hilda Orășan, Vasile Negrean, Lorena Ciumarnean, Doina Adina Todea

**Affiliations:** 15th Departament Internal Medicine, 4th Medical Clinic, University of Medicine and Pharmacy, 400015 Cluj-Napoca, Romania; Teodora.Alexescu@umfcluj.ro (T.-G.A.); Olga.Orasan@umfcluj.ro (O.H.O.); Vasile.Negrean@umfcluj.ro (V.N.); Lorena.Ciumarnean@umfcluj.ro (L.C.); 2Clinical Rehabilitation Hospital, 400347 Cluj-Napoca, Romania; madalinakisanovici@yahoo.ro; 3Department of Pneumology, University of Medicine and Pharmacy, 400332 Cluj-Napoca, Romania; Lupascu.Anca@umfcluj.ro (A.-D.M.); dtodea@umfcluj.ro (D.A.T.); 4SC Sanprev SRL, 400202 Cluj-Napoca, Romania; sanprev@yahoo.com; 5Department of Informatics and Biostatistics, University of Medicine and Pharmacy, 400012 Cluj-Napoca, Romania; dleucuta@umfcluj.ro

**Keywords:** COVID-19 pandemic, mental health, neuroticism state, pre-university education, lockdown

## Abstract

*Background and objectives**:* Since the first reports of SARS-CoV-2 infection cases in China, the virus has rapidly spread to many countries, including Romania. In Romania, schools were closed in March 2020 to prevent the virus from spreading; since then, they have been sporadically opened, but only for a short time. Teachers had to adopt online education methods, experiencing real difficulties in their attempts to maintain high-quality teaching, as a result of social distancing from students and colleagues. The current study aimed to evaluate the burden on the neuroticism states of employees in the pre-university education system during the COVID-19 pandemic. *Materials and Methods*: A prospective study was conducted, in which personality trait data from 138 employees were collected via a questionnaire (EPI, Eysenck Personality Inventory), which measured extraversion–introversion and neuroticism scores before and during the COVID-19 pandemic. Initially, 150 subjects were invited to participate in the study, although 12 of them refused to participate. Based on the questionnaire not being fully filled in a further three subjects were excluded from the study, leaving a total of 135, of which 115 were woman and 20 were men. *Results*: The results demonstrate that the subjects included in the study expressed higher neuroticism during the COVID-19 pandemic than in the pre-pandemic period. This change could promote more stress and depression symptoms. Subjects with high school education had significantly lower neuroticism scores over time than those with university education (*p* = 0.006). Furthermore, we found extraversion scores to be statistically significant in our population (*p* = 0.022). *Conclusion*: The gender and living environment of the teachers were not significantly associated with the reduction in the extraversion score, but were more frequently found among older persons and in subjects without higher education. Subjects of Hungarian ethnicity had lower extraversion scores than those of Romanian ethnicity.

## 1. Introduction

The global SARS-CoV-2 pandemic originated in Wuhan City, Hubei province, China, and has spread rapidly to become a major public health problem worldwide [1,2], leading to extensive and devastating social and economic impacts. During this crisis, authorities have taken numerous measures, such as the cessation of social and economic activities for a specific time [3]; lockdown in Romania began in March 2020. The lockdown in Romania spanned a period of 2 months, from March to May, and affected various economic sectors, such as airline companies, hotels, restaurants, factories, etc. [4]. All around the world, hundreds of thousands of people lost their jobs or reduced their activity. The impact of lockdown on pre-university teachers translated into a shift from traditional education and examination methods to online methods. Many teachers experienced difficulties in teaching online, because only part of them possessed the required technical and digital skills. Many faced a higher volume of work than before the pandemic, and distancing from their colleagues and students led to mental health problems in some cases. Given the contagious nature and the prolonged incubation period of the virus (2 to 14 days, with a mean of 5–6 days), many teachers feared they might infect their family members, friends, or colleagues [1,5]. All these restrictive measures undoubtedly affected the social and mental health of teachers, and people in general, including changes in neuroticism state. In addition, some studies have shown the prevalence of PTSD (post-traumatic stress disease) ranging from 7% to 53.8% during a pandemic [6,7]. The rapidly expanding mass hysteria and panic surrounding COVID-19 can cause lasting psychological problems, which could be even more harmful in the long term than the virus itself [8]. Despite attempts to live with minimal exposure to the virus, anxiety, and the difficulties of adapting to a new living environment, led to major health problems, such as depression. Stress, anxiety, and/or depression caused a decrease in the productivity of teachers, increased their chances of making inadequate decisions, and affected their personal life. For most of them, the COVID-19 pandemic meant a radical lifestyle change, in perceptions about life in general, and a reconsideration of personal values; this can be managed with the help of psychological support groups made available to employees [1,9].

Currently, in Romania, it is still declared the state of health “alert”, which means, for pre-university education, that the learning and teaching process is carried out in a hybrid system (onsite and online), the conditions being established every month, meaning a lack of predictability for both children and teachers.

While countries are struggling with the fourth wave, continued isolation due to blockages and the fear that one of their loved ones will contract COVID-19 are just some of the reasons that continue to cause an unprecedented psychological burden. Almost two years after the first case of COVID-19, attempts were made to highlight the long-term psychological and cognitive consequences of the pandemic, not only from SARS-CoV-2 infection, but also from major psychosocial stress due to prevailing conditions, increasing mortality, and the distancing mechanisms imposed [10].

Lockdown can be associated with mood disorders, depression, stress, fear, insomnia, nervousness, and irritability. The fear generated by the quarantine lead to anxiety about financial problems, and worries about prolonged time spent in front of computers [11]. However, after the lifting of restrictions, quality of life should improve, and the levels of stress and anxiety should reduce [12].

The mental health of an individual can be analyzed by certain indicators. Neuroticism designates general emotional overreaction and a predisposition to depression under the effect of stress. Neuroticism is generally measured using self-report questionnaires as part of a personality assessment. It is also very important to emphasize that neuroticism is a dimension and not a diagnosis. Both positive (extraversion) and negative (intraversion) outcomes of neuroticism have been observed. Neuroticism may lead to secondary mental health issues, such as depression or anxiety, and implicitly, additional medical costs, both for the individual and for society in general, absence from work, low performance at work [13]. It has long been established as one of the more important and significant domains of personality, and is becoming increasingly recognized as a fundamental domain of personality disorders, and psychopathology more generally [11,14].

Neuroticism is often confused with neuroses. In basic terms, neurosis is a disorder involving obsessive thoughts or anxiety, while neuroticism is a personality trait that does not have the same negative impact on everyday living. It is generally measured using self-report questionnaires as part of a personality assessment. It might also involve asking other people, such as friends and family, who know the individual well, about their personality characteristics [13].

Here, we mention some elementary concepts, allowing us to communicate the aim of this study. Extraversion, as opposed to introversion, designates the externalization and non-inhibition tendencies, and the impulsive and social tendencies, of an individual. The results are interpreted depending on the three scales, E (extraversion), N (neuroticism), and L (lie) [15].

Ambiversion refers to those who possess both extroversion and introversion. Interestingly, ambiverts can have introverted characteristics, such as listening or introspection, while also exhibiting extroverted characteristics, such as enthusiasm [14,15].

The current study mainly aimed to evaluate whether employees in the pre-university education sector during the COVID-19 pandemic presented changes in personality compared to the pre-pandemic period, more precisely, changes in neuroticism and extraversion states, respectively.

The secondary objectives were to establish which types of neuroticism change were presented by the subjects, and whether there exists a difference between the subjects studied in terms of age, gender, ethnicity, and educational level.

## 2. Materials and Methods

### 2.1. Study Design and Setting

To evaluate personality changes from before to during the pandemic, the Eysenck Personality Inventory (EPI), validated in 1964 [12,13], was used in a prospective study on 138 subjects working in the field of pre-university education. Data collection was initiated in May 2018, within the occupational medicine program, through SC. SANPREV SRL, by each subject completing neuroticism questionnaires, having been allocated at least 20 min. The questionnaires were physically completed each time, both pre-pandemic and after the onset of the COVID-19 pandemic by scheduling subjects at equal intervals necessary to assess health, according to current legislation on occupational medicine in Romania. The design of the study, the interpretation, and data collection were carried out in collaboration with the C.F Cluj-Napoca Clinical Hospital, where the approval of the Ethics Committee was also received. The first inventory was performed to evaluate the personality of pre-university education employees. It was repeated in June 2020, under the same conditions of the occupational medicine program, after the lockdown period in Romania (15 March–15 May 2020), which was a stressful period for the entire population. As in the first assessment, the second assessment immediately after the lockdown period was physical, in compliance with all the norms of sanitary protection.

### 2.2. Variables

Demographic data, including age, gender, ethnicity, education (≤10 grades, vocational school, high school, post-secondary school, university), and place of residence, were collected along with the answers to the Eysenck Personality Inventory (extraversion–introversion, neuroticism–stability, and lie scales—each one measured as scores and transformed into categories based on the literature thresholds).

### 2.3. Data Sources

The Eysenck Personality Inventory was elaborated by Hans J. Eysenck, and assesses human personality based on two general dimensions, independent of one another: extraversion–introversion and neuroticism–stability [15,16,17,18]. The EPI is composed of two parallel questionnaires, for cases in which retesting is required, each consisting of 57 questions that can be answered in 10 min, and are grouped into three scales: the extraversion–introversion scale (E), the neuroticism scale (N), and the lie scale (L). The E scale highlights the tendency towards extraversion or introversion, and has limits between 0 and 24 (0–10 = introvert subjects, 10–16 = ambivert subjects, 16–24 = extrovert subjects). The typical extrovert is a sociable, impulsive, carefree, and optimistic individual, while the typical introvert is a quiet, reserved, distant, and slightly pessimistic individual, with a tendency to develop symptoms of anxiety and depression. The N scale highlights emotional stability, and has limits between 0 and 24 (0–13 = low level of neuroticism, 14–24 = high level of neuroticism. Individuals who have high N grades tend to be emotional, hyperreactive, and prone to anxiety and depression. The L scale a set of items within a psychological instrument (particularly a personality assessment) used to indicate whether a respondent has been truthful in answering, and allows the identification of the degree of desirability of the subject’s answers. This type of scale utilizes redundant questions, which are questions that seek to elicit the same information, but use varied question formats to check the consistency of the answers. In the EPI questionnaire, results between 0 and 5 show that the test can be considered valid, and results between 6 and 9 invalidate the test results [15,16,17,18].

The questionnaire was self-administered, the instructions for the administration of the EPI were written in full on each questionnaire notebook, and the questions were read and solved independently by each subject. This questionnaire comprises questions referring to the behavior of the subjects, the way in which they act in stressful situations, and certain feelings generated by stressful situations. The answer YES or NO is marked in the check boxes corresponding to the questions. As the questions are written in an accessible language, the EPI does not pose difficulties in understanding them, and thus, they can be applied to all subjects—older than 18 years of age—regardless of their social and cultural living or working environment. Given the relatively small number of questions, there is no risk of the subjects becoming tired or bored. The items of the EPI questionnaire require choosing between two-answer possibilities. This limits the subject’s freedom of expression, but facilitates rating the answers.

### 2.4. Statistical Analysis

Categorical data were presented as counts and percentages. Continuous, normally distributed data were presented as means (standard deviation), while continuous, non-normally distributed data were presented as medians and quartiles.

To compare the pre-COVID-19 and COVID-19 scores, we used several paired statistical tests: the McNemar test for binary data, the Stuart–Maxwell test for nominal data, and the Wilcoxon signed-rank test for data that were not normally distributed.

To assess the differences between the pre-COVID-19 and COVID-19 scores, we performed univariate regression followed by multiple linear regression, using the sandwich estimator for coefficient standard errors. The models predicted the differences in scores, and used independent variables (age, gender, living environment, education, and ethnicity) as predictors. We checked the normality of residuals, the presence of multicollinearity (with variance inflation factors), heteroscedasticity (with the Breusch–Pagan test), and the functional form (using component + residual plots). The regression coefficients, along with 95% confidence intervals and *p*-values, are presented in the following section. For all statistical tests, we used a significance level alpha of 0.05, and two-tailed *p*-values were computed. All statistical analyses were carried out in the R environment for statistical computing and graphics (R Foundation for Statistical Computing, Vienna, Austria), version 4.0.2 [19].

## 3. Results

Participation in the study was proposed to a total of 150 people, comprising teachers and auxiliary subjects from the pre-university sector, more specifically, from a single high school in Cluj-Napoca. The study inclusion criteria were that subjects must be professionally active when filling in the questionnaire, and the minimum age for participation was 18 years. The exclusion criterion was the failure to answer at least one question of the questionnaire before and during the pandemic. Following the inclusion and exclusion criteria, 135 complete questionnaires remained to be studied (Figure 1).

All subjects gave their informed consent for inclusion before participating in the study. The study was carried out in accordance with the declaration from Helsinki, and the protocol was approved by the Ethics Committee of the C.F Cluj-Napoca Clinical Hospital, with the approval 2A/30.03.2020.

The overall response rate of the participants was 97.82%. The subjects were aged between 29 and 66 years, with a mean age of 47.87 years (±8.21). From the total number of participants, 115 were female (85.19%) and only 20 were male. The data regarding living environment, level of education, and ethnicity are presented in Table 1.

There was a statistically significant difference between the values before and during the COVID-19 pandemic regarding the median scores of extraversions and neuroticism, which diminished by one point (Table 2).

Of the 109 subjects with ambiversion in the pre-COVID-19 period, 24 changed to introversion and none to extraversion. The two subjects with extraversion in the pre-COVID-19 period changed to ambiversion during the COVID-19 pandemic. Of the 24 subjects with introversion in the pre-COVID-19 period, 10 changed to ambiversion during the COVID-19 pandemic, and none to extraversion (Table 3). The lie score values remained similar, but, when comparing the dichotomized lie score, there were statistically significant differences over time.

### 3.1. Score Evolution over Time

#### 3.1.1. Extraversion Scores

The difference between initial and final extraversion scores were computed using simple and multiple linear regression models (Table 4). The multivariate regression model had a significance level of *p* = 0.022 (F = 2.58, with 6.128 d.f. (degrees of freedom), adjusted determination coefficient = 0.07). Higher age was significantly associated with lower extraversion scores (*p* = 0.048). Subjects with high school education had significantly higher extraversion scores over time than those with university education (*p* = 0.002). Romanian ethnicity subjects had significantly higher extraversion scores in time than those of Hungarian ethnicity (*p* = 0.021). Gender and living environment were not significantly associated with extraversion score reduction.

#### 3.1.2. Neuroticism Score

The difference between initial and final neuroticism scores were computed using simple and multiple linear regression models (Table 5). The multivariate regression model had a significance level of *p* = 0.033 (F = 2.58, with 6, 128 d.f., adjusted determination coefficient = 0.06). Subjects with high school education had significantly lower neuroticism scores over time than those with university education (*p* = 0.006), both in univariate and multivariate models. Age (*p* = 0.303), gender (*p* = 0.376), living environment (*p* = 0.351), and ethnicity (*p* = 0.531) were not significantly associated with neuroticism score reduction.

#### 3.1.3. Lie Scale

The difference between final and initial scores were computed using simple and multiple linear regression models (Table 6). The multivariate regression model had a significance level of *p* = 0.472 (F = 0.94, with 6, 128 d.f., adjusted determination coefficient = 0.06). No statistically significant independent predictors for the difference in scores were found.

## 4. Discussion

In our study, it was demonstrated that subjects presented stress and depression related to their workplace during the COVID-19 pandemic, measured by neuroticism score. Additionally, in subjects that were older, without higher education, and of Hungarian ethnicity, the extraversion score was reduced.

Previous research has found neuroticism to be a risk factor for suicide among men and women in the general UK population. Neurotic individuals also tend to negatively interpret social interactions, and may lash out angrily in response to perceived criticism and rejection [20].

Stress at the workplace is a negative physical and/or emotional response that occurs when the workplace requirements do not correspond to the employee’s abilities, resources, and/or needs. It has been demonstrated that more than half of employees are subjected to intense stress, and two-thirds of them experience difficulties focusing in the workplace because of stress. More precisely, stress at the workplace leads to negative reactions and signs of tension, such as physical and mental fatigue, which may induce additional pathological conditions, such as obesity, hypertension, diabetes mellitus, sleep disorders, anorexia, and muscle stiffness; when stress-related symptoms worsen, chronic stress develops. In addition, stress can affect performance at the workplace and lead to the exacerbation of the mental health problems, potentially causing mental disorders, as well as changes in personality traits, such as depression [19,20,21,22].

The COVID-19 pandemic caused episodes of depression, anxiety, and/or sleep disorders in the general population. The cause of these disorders is probably multifactorial, as shown by Deng et al. [23].

It should be noticed that older teachers reported the highest levels of anxiety and stress (*p* = 0.048), highlighting the greater adaptability of young people to online teaching conditions. However, even young teachers experienced high levels of stress, typically because of the greater instability in the workplace, although this has not been demonstrated in our study [24]. A study from the United Arab Emirates also found that older employees in different fields developed mental health problems, such as depression, more often due to their high mortality rate due to COVID-19, which makes them physically and mentally vulnerable. In general, these were single people with low social support, and with limited access to online mental health services due to lack of technological skills [25].

Sex was not associated with changes in extraversion and neuroticism scores in our study. In contrast, studies in China, the United Arab Emirates, and Italy during the pandemic have shown that women are more prone to depression than men, and have a greater psychological vulnerability to stress, suggesting that they may respond more intensely than men in the event of a pandemic [25,26,27].

Racial and ethnic minorities have significantly lower access to stable housing, decent wages, and physical and mental health resources then the majority groups, which poses a higher risk of negative consequences on mental health and personality traits. It was demonstrated that persons of lower socioeconomic status have higher rates of depression and anxiety symptoms, changes in personality traits, after events such as the COVID-19 pandemic, compared to those of higher socioeconomic status [28]. Similarly, the academic achievements of pupils and students without resources are frequently affected in the context of remote learning because of the limited capacity to acquire the necessary technology. In our study, socioeconomic status was not investigated; in contrast, there was a considerable ethnic difference, as 60% of the subjects were of Hungarian ethnicity. Unlike Romanian subjects, these subjects had a lower extraversion score (*p* = 0.021).

Teachers were exposed to major stress, as our study also demonstrates, because of the need to adapt in a very short time to online teaching methods, as well as due to the lack of equipment for both teachers and students, particularly those of disadvantaged groups and/or from rural areas. The IRES study [29] aimed to find out how many children in Romania have access to online education and to the necessary infrastructure. Only 62% of pupils in rural areas are equipped from this point of view, compared to their peers from urban areas (76%). Another study showed that many teachers experienced difficulties, because only part of them had sufficient technical and digital skills for this type of activity—online teaching. It was found that working at home using information and communication technologies may create feelings of tension, anxiety, exhaustion, and a decrease in satisfaction in the workplace. Many teachers experienced a higher volume of work, and distancing from their colleagues and students generated serious mental health problems [30]. Teachers working in pre-school and primary education feel more responsible for younger children, who need more care and protection due to their age [24,30].

Another main risk factor in developing depression or anxiety is the lack of contact with the family and friends during the lockdown imposed to prevent the risk of spreading SARS-CoV-2. People who had fewer social interactions during the pandemic were found to experience significant detrimental impacts on cognitive function [23,31]. Additionally, the associated comorbidities lead to the fear of COVID-19 disease, and raise the risk of developing several symptoms of anxiety and stress [32,33,34,35]. Anxiety stems, in part, from their compromised immunity caused by pre-existing conditions, making them susceptible to infection and at a higher risk of mortality [22,34].

The high level of confidence in governmental actions to cope with COVID-19 challenges, and an extremely subjective level of knowledge regarding COVID-19, are associated with a low burden on mental health; this was not predictable in the field of education in Romania, given that the decisions to completely or partly shift to online education were made on short notice, Romanian education before the COVID-19 pandemic being exclusively organized on-site and with reduced digital infrastructure [35,36].

Another factor that caused anxiety and depression among teachers was the absence of physical activity. Teachers who spend more time performing physical activities have better teaching performance, as well as better relationships in their personal lives and with their students, a fact demonstrated by a study in Spain; however, in our study, level of physical activity was not evaluated. Indoor exercise could be a potential therapy not only to maintain robust physical health, but also to reduce the psychological impact during this trial period [37,38].

Employees working in service industries (for example, airline companies, hotels, restaurants) may experience extreme stress because of the direct and indirect contact with various clients of different nationalities, working in closed spaces, and the management of articles used by various persons [19,39]. Although this has not been studied, the same risks can be found among teachers who carry out their activity on a hybrid basis, or even on-site. By directly working with pupils and students, and by changing classes every hour, these individuals are exposed to a higher risk of contracting the virus, and consequently, they experience increased level of stress and anxiety.

Studies from Romania show a significant difference between workers in healthcare and social fields and workers in administration and management areas. Medical staff seem to be the most affected during this period. There are several specific sources of anxiety and stress related to the COVID-19 pandemic among the medical staff, such as concerns about the availability of personal protective equipment, exposure to COVID-19 at the workplace, infection of their family members, and the uncertainty that their organization will take care of their personal needs if they become infected. In addition, the lack of support from other persons, and concerns about their families as workload increases, the desire to provide competent medical care, and the fear of making errors also contributes to stress and anxiety [40]. Another problem is the fact that, in addition to their hospital activity, some doctors are also involved in teaching activities. These individuals had, in turn, to reinvent themselves, and rapidly adapt to the online or hybrid teaching methods, to maintain the quality of medical education. These sources do not fit in the usual scenario of the workplace, leading to exhaustion as well as post-traumatic stress disorder [41]. Furthermore, it should be considered that the education received by young people at this time of crisis will form the society of the future. Consequently, if a high standard of education is desired, the psychological well-being of teachers who provide it should be protected.

Possible actions to mitigate the impact of the pandemic on changes in personality traits include an improvement of the workplace infrastructure, special pedagogical courses for online or hybrid teaching, mental health and personality monitoring, and understanding the various personal needs of the teaching staff. Setting up mental health organizations specific to future pandemics, with branches across many nations and individual healthcare institutions for research, mental healthcare delivery, and raising awareness at both personal and community levels is desperately needed. Structured websites and toll-free helplines may be launched for alleviating psychological distress among the public regarding this ongoing pandemic. Social media should be used in a sensible manner to educate people on transmission dynamics, symptoms of disease, and exact times when medical consultations are needed. To protect social media from devaluations, strict government laws and legislation regarding fake news, social media rumors, disinformation, and misinformation are to be implemented [42].

The strength of this study is represented by the fact it is, to our knowledge, the only study to prove the change in the mental state of pre-university education workers during the COVID-19 pandemic. Moreover, this study compared pre-COVID-19 data with COVID-19 data in a prospective design, and had significant results.

There are some limitations of this study. First, being an observational study, confounding can be an important issue. The before–during design limits the internal selection bias, but it is difficult to account for secular trends, regression to the mean, and other external influences on the study participants. Next, we analyzed workers in only one education institution, and the connection of employees with family/friends during the lockdown, the effects of lack of physical activity during this period and pre-existing conditions were not studied. Another limit is that the questionnaire was self-administered, although all subjects were advised to ask for help if they had difficulty understanding the questions. The gender limitation exists because females stated higher scores on the scale measuring neuroticism, and had a higher fear of COVID-19 than males [43].

## 5. Conclusions

There was a statistically significant difference between the values before and during the COVID-19 pandemic, regarding the median scores of neuroticisms and extraversion among pre-university education employees during the COVID-19 pandemic.

Regarding the neuroticism score, this was modified only in the case of subjects without higher education, while age, sex, living environment, and ethnicity were not associated with score changes.

The extraversion score was more frequently found lower among older persons and in subjects without higher education, as well as in subjects with Hungarian ethnicity.

The COVID-19 pandemic has brought not only medical implications, but also changes in personality traits among employees in pre-university education. These results can be used to guide employees who need counseling to seek advice from a psychologist. This could help people who encounter emotional concerns, social, vocational, developmental, and organizational problems, and educational and health-related issues to overcome these difficulties in their lives via interventions, such as the Unified Protocol (a form of Cognitive Behavioral Therapy), mindfulness, and behavioral interventions to help them learn to regulate their emotions [44,45].

## Figures and Tables

**Figure 1 medicina-58-00895-f001:**
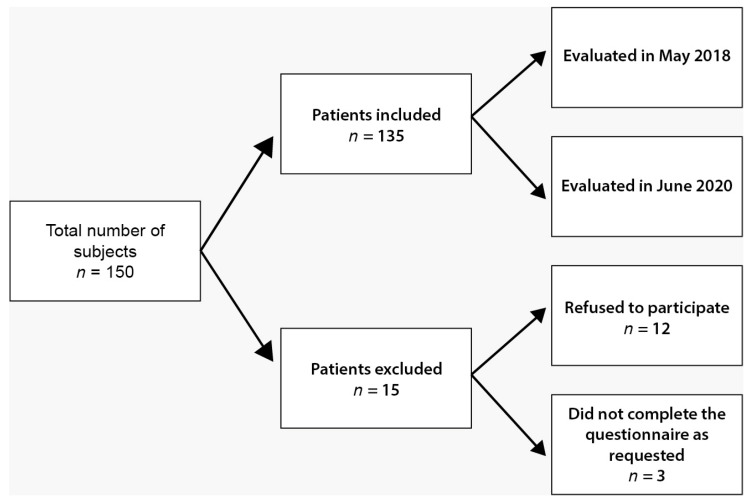
Study flowchart regarding inclusion and exclusion criteria.

**Table 1 medicina-58-00895-t001:** Demographic characteristics of the studied population.

Characteristic	Number (%)
	(*n* = 135)
Age (years), mean (SD)	47.87 (8.21)
Gender (female/total)	115/135 (85.19)
Living environment (urban/total)	130/135 (96.3)
Education	≤10 grades: 4/135 (2.96)
	Vocational school: 2/135 (1.48)
	High school: 20/135 (14.81)
	Post-secondary school: 5/135 (3.7)
	University: 104/135 (77.04)
Ethnicity (Romanian/total)	53/135 (39.26)

**Table 2 medicina-58-00895-t002:** Comparison of Eysenck Personality Inventory in the pre-COVID-19 versus the COVID-19 period.

			Pre-COVID-19		
		Introversion	Ambiversion	Extraversion	*p*-Value
		(*n* = 24)	(*n* = 109)	(*n* = 2)	
				0.016
COVID-19	Introversion, n (%)	14 (58.33)	24 (22.02)	0 (0)	
	Ambiversion, n (%)	10 (41.67)	85 (77.98)	2 (100)	
	Extraversion, n (%)	0 (0)	0 (0)	0 (0)	

**Table 3 medicina-58-00895-t003:** Comparison of Eysenck Personality Inventory in the pre-COVID-19 versus the COVID-19 period.

Time:	Pre-COVID-19	COVID-19	Difference	*p*-Value
			(95% CI)	
Extraversion score, median (IQR)	11 (10–12)	10 (9–11)	1 (0.5–1.5)	<0.001
Extraversion scale				
Introversion	24 (17.78)	38 (28.15)		
Ambiversion	109 (80.74)	97 (71.85)		
Extraversion	2 (1.48)	0 (0)		
Neuroticism score, median (IQR)	3 (1–5)	2 (1–4)	1 (0.5–1.5)	0.001
Neuroticism scale				
Emotional instability	4 (2.96)	0 (0)	4 (2.96)	<0.001
Emotional stability	131 (97.04)	135 (100)		
Lie score, median (IQR)	5 (4–6)	5 (4–6)	0 (−0.5–0)	0.122
Lie scale				0.03
Valid test	86 (63.7)	74 (54.81)		
Invalid test	49 (36.3)	61 (45.19)		

CI, confidence interval.

**Table 4 medicina-58-00895-t004:** Simple and multiple linear regression models predicting extraversion score differences (pre-COVID-19 versus COVID-19 period).

	B	(95% CI)	*p*-Value	B	(95% CI)	*p*-Value
	Unadjusted			Adjusted		
Age (years)	0	(−0.04–0.04)	0.989	0.05	(0–0.1)	0.048
Gender (male vs. female)	−0.18	(−0.94–0.58)	0.647	−0.44	(−1.22–0.33)	0.263
Living environment (urban vs. rural)	0.05	(−2.09–2.2)	0.961	−0.53	(−2.29–1.23)	0.556
Education (high school vs. university)	−1.4	(−2.29–−0.5)	0.003	−1.57	(−2.54–−0.61)	0.002
Education (other vs. university)	−0.46	(−1.5–0.58)	0.388	−0.82	(−1.75–0.11)	0.086
Ethnicity (Romanian vs. Hungarian)	−0.69	(−1.36–−0.01)	0.048	−0.91	(−1.67–−0.14)	0.021

CI, confidence interval; B, unstandardized beta.

**Table 5 medicina-58-00895-t005:** Simple and multiple linear regression models predicting neuroticism score differences (pre-COVID-19 versus COVID-19 period).

	B	(95% CI)	*p*-Value	B	(95% CI)	*p*-Value
	Unadjusted			Adjusted		
Age (years)	0.01	(−0.06–0.08)	0.848	−0.04	(−0.12–0.04)	0.303
Gender (male vs. female)	−0.69	(−1.95–0.57)	0.288	−0.5	(−1.61–0.61)	0.376
Living environment (urban vs. rural)	1.02	(−1.06–3.11)	0.338	1.21	(−1.33–3.75)	0.351
Education (high school vs. university)	2.57	(0.81–4.34)	0.005	2.71	(0.81–4.61)	0.006
Education (other vs. university)	−0.26	(−2.1–1.58)	0.781	0.01	(−1.86–1.88)	0.994
Ethnicity (Romanian vs. Hungarian)	0.37	(−0.72–1.45)	0.51	0.39	(−0.83–1.6)	0.531

CI, confidence interval; B, unstandardized beta.

**Table 6 medicina-58-00895-t006:** Simple and multiple linear regression models predicting lie score differences (pre-COVID-19 versus COVID-19 period).

	B	(95% CI)	*p*-Value	B	(95% CI)	*p*-Value
	Unadjusted			Adjusted		
Age (years)	0	(−0.04–0.04)	0.943	0	(−0.04–0.05)	0.886
Gender (male vs. female)	−0.19	(−1.07–0.7)	0.679	−0.13	(−0.97–0.72)	0.766
Living environment (urban vs. rural)	0.06	(−0.41–0.53)	0.798	0.21	(−0.7–1.13)	0.646
Education (high school vs. university)	0.42	(−0.48–1.31)	0.366	0.48	(−0.45–1.41)	0.313
Education (other vs. university)	0.77	(−0.42–1.97)	0.207	0.77	(−0.55–2.09)	0.254
Ethnicity (Romanian vs. Hungarian)	−0.49	(−1.1–0.12)	0.12	−0.56	(−1.26–0.15)	0.126

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
