# Peer review of "Change in Neuroticism and Extraversion among Pre-University Education Employees during the COVID-19 Pandemic"

_medicina, 2022, doi:10.3390/medicina58070895_

Round 1

Reviewer 1 Report

Thank you for the opportunity to review this work. I have some suggestions to improve your work:

Abstract
Please be more clear in the result part of the abstract section (it is a little bit confusing)

Introduction
Please try to summarize the first part of the introduction related to COVID-19 (42-54).

It is not a common thing a change in personality, usually it is related to specific traumatic experiences. Please insert in the introduction section why you think that a change in personality happened after the covid-19 event (with the related literature).

The reference [12] has a link that does not work

You talk about neuroticism citing the Big 5 and the Eysenck Personality Inventory. However, there are many differences between these two conceptions of neuroticism. As you used the EPI, I suggest you to delete the Big5's part and focus on the EPT conception of neuroticism (87-100)

Please add the correct reference of EPI (if you used the Romanian version, cite the Romanian validation) ...

Study design and setting
How did you present this research to participants? How did you make the recruitment?

Participants
In the inclusion criteria of your study, what do you mean by "personal desire to complete the questionnaire"? I think it is not clear and not necessary... In the same way, their refusal to participate in the study is not an exclusion criterion...

Results
There is a typo in line 212, SD is the Standard deviation (8.21)

Table 2 is not clear, please reformulate it in its contents. 

The concept of AMBIVERSION is not defined in the precedent sections, please provide an extensive explanation of the EPI.

Discussion
Please focus the discussion on the variable you explored and try to explain your results in a more precise and linear way. Try to make this section short and focused on the variables you considered.

Limitations
Your sample is not well balanced between males/females.

Conclusions
Please try to explain how these results could be useful for the employees (e.g. specific psychological support...). 

Author Response

Comments and Suggestions for Authors

Thank you for the opportunity to review this work. I have some suggestions to improve your work:

We are glad for these valuable comments to our work that helped us to improve our paper.

Abstract

Please be more clear in the result part of the abstract section (it is a little bit confusing)

Thank you for this advice, we rewrote the results in the abstract section beginning with the line no.29

Introduction

Please try to summarize the first part of the introduction related to COVID-19 (42-54).

Thank you for your suggestion. Now we have modified the introduction section and it can be found in the first paragraph of the section.

It is not a common thing a change in personality, usually it is related to specific traumatic experiences. Please insert in the introduction section why you think that a change in personality happened after the covid-19 event (with the related literature).

Thank you for the question. The authors expressed their opinion according to the literature on lines 116-120.

The reference [12] has a link that does not work

Now the reference 12 is reference 14

You talk about neuroticism citing the Big 5 and the Eysenck Personality Inventory. However, there are many differences between these two conceptions of neuroticism. As you used the EPI, I suggest you delete the Big5's part and focus on the EPT conception of neuroticism (87-100)

Thank you very much for your valuable suggestion. We have deleted the part with the Big 5

Please add the correct reference of EPI (if you used the Romanian version, cite the Romanian validation) ...

Reference number 14 is illustrating the Romanian version of the EPI questionnaire.

Study design and setting

How did you present this research to participants? How did you make the recruitment?

Thank you very much for the question. We presented the objective of the study: we wanted to develop a study among teachers and auxiliary personnel from the pre-university sector in Romania to see how the pandemic affect their mental health and the eventual changes in their personality traits. It was an interesting option

The participants gave their consent to be involved in this study as they completed this questionnaire during a psychological evaluation (led by an authorised clinical psychologist), initiated in May 2020, which was included in the regular Labour Medicine examination for teachers and auxiliary personnel working in pre-university education.

Participants

In the inclusion criteria of your study, what do you mean by "personal desire to complete the questionnaire"? I think it is not clear and not necessary

Thank you for your mention. We have deleted it.

In the same way, their refusal to participate in the study is not an exclusion criterion...

Thank you for the observation. We have modified the manuscript now.

Results

There is a typo in line 212, SD is the Standard deviation (8.21)

Thank you for this remark, we have deleted the SD.

Table 2 is not clear, please reformulate it in its contents.

Dear Reviewer, we have changed the content in order to be better understood.

The concept of AMBIVERSION is not defined in the precedent sections, please provide an extensive explanation of the EPI.

Thank you for this mention. We have defined now the ambiversion concept in the introduction section: ‘’Ambiversion refers to those who possesses both extroversion and introversion. Interestingly, the ambiverts can have introverted characteristics like listening or introspection and can borrow extroverted characteristics like enthusiasm [15,16]’’.

Discussion

Please focus the discussion on the variable you explored and try to explain your results in a more precise and linear way. Try to make this section short and focused on the variables you considered.

We have modified the discussion section.

Limitations

Your sample is not well balanced between males/females.

I know this is a limit to our study, but these subjects agreed to take part in our study and completed a valid questionnaire. We added this gender criterion to the limitations in lines 441-443.

Conclusions

Please try to explain how these results could be useful for the employees (e.g. specific psychological support...).

Thank you for your suggestion. We have modified the conclusion section to highlight the advice for employees facing these psychological issues.

Reviewer 2 Report

I would like to thank the authors for the opportunity to review this article. 

In my opinion, the work has potential however it needs some corrections. First of all, the introduction is, in my opinion, too long and contains unnecessary information, e.g. the initial paragraph - what really adds to the point of the article. 

Moreover, in my opinion citations are missing in many places. It should be supplemented. 

Methodological description does not tell us explicitly what type of study it is, was exactly the same group of teachers examined? 

The description of the studied group should be in the results section.

What I miss in the discussion is a kind of summary that would explain what gap in the literature your study fills. Does it have any practical implications? If so, which ones.

Author Response

Comments and Suggestions for Authors

I would like to thank the authors for the opportunity to review this article.

We are glad for these valuable comments to our work that helped us to improve our paper.

Dear Reviewer,

Thank you very much for your valuable comments that helps us to improve our work.

In my opinion, the work has potential however it needs some corrections. First of all, the introduction is, in my opinion, too long and contains unnecessary information, e.g. the initial paragraph - what really adds to the point of the article.

Thank you for this suggestion. We have modified the introduction and it can be now found in the new version of the manuscript.

Methodological description does not tell us explicitly what type of study it is, was exactly the same group of teachers examined?

Thank you for the question.

It is a prospective study that included preuniversity teachers and auxiliary personnel. The questionnaires were applied to the exact same people and if one missed one session, it was excluded from the study.

The description of the studied group should be in the results section

Thank you for your advice, we have modified this.

What I miss in the discussion is a kind of summary that would explain what gap in the literature your study fills. Does it have any practical implications? If so, which ones.

We wanted to study a particular population, pre-university teachers and auxiliary personnel in Romania and the impact of the COVID-19 pandemic on their neuroticism and extraversion score.
